# Longitudinal Effects of Lipid-Lowering Treatment on High-Risk Plaque Features and Pericoronary Adipose Tissue Attenuation Using Serial Coronary Computed Tomography

**DOI:** 10.3390/diagnostics15182340

**Published:** 2025-09-16

**Authors:** Loris Weichsel, Florian André, Matthias Renker, Lukas D. Weberling, Philipp Breitbart, Daniel Overhoff, Meinrad Beer, Borbála Vattay, Sebastian Buss, Mohamed Marwan, Stefan Baumann, Andreas A. Giannopoulos, Natalia Solowjowa, Sebastian Kelle, Norbert Frey, Grigorios Korosoglou

**Affiliations:** 1Vascular Medicine & Pneumology, Cardiology, GRN Hospital Weinheim, 69469 Weinheim, Germany; 2Cardiac Imaging Center Weinheim, Hector Foundations, 69469 Weinheim, Germany; 3Cardiology, Angiology & Pneumology, University Hospital Heidelberg, 69120 Heidelberg, Germany; 4DZHK (German Centre for Cardiovascular Research), Partner Site Heidelberg/Mannheim, 69120 Heidelberg, Germany; 5Department of Cardiology, Campus Kerckhoff of the Justus Liebig University Giessen, 35390 Bad Nauheim, Germany; 6German Centre for Cardiovascular Research (DZHK), Partner Site Rhein Main, 61231 Bad Nauheim, Germany; 7Department of Cardiology and Angiology, Medical Center–Faculty of Medicine, University of Freiburg, 79098 Bad Krozingen, Germany; 8Department of Radiology and Nuclear Medicine, University Medical Center Mannheim (UMM), Heidelberg University, 68167 Heidelberg, Germany; 9Department for Diagnostic and Interventional Radiology, University Hospital Ulm, 89081 Ulm, Germany; 10Heart and Vascular Center, Semmelweis University, 1085 Budapest, Hungary; 11MVZ-DRZ Radiology Center, 69126 Heidelberg, Germany; 12Department of Cardiology, University of Erlangen, 91054 Erlangen, Germany; 13Department of Cardiology, District Hospital Bergstraße, 64646 Heppenheim, Germany; 14Department of Nuclear Medicine, Cardiac Imaging, University Hospital Zurich, 8091 Zurich, Switzerland; 15German Heart Center Berlin, 13353 Berlin, Germany; 16DZHK (German Centre for Cardiovascular Research), Partner Site Berlin, 10785 Berlin, Germany

**Keywords:** coronary artery disease, serial evaluation of high-risk plaque features, low/high intensity lipid-lowering treatment, non-calcified plaque, pericoronary adipose tissue attenuation (PCAT), multi-center study

## Abstract

**Aim**: To evaluate the impact of different lipid-lowering treatment intensities on high-risk plaque features and pericoronary adipose tissue (PCAT) attenuation in patients undergoing serial coronary computed tomography angiography (CCTA). **Methods**: Individuals with suspected or known coronary artery disease (CAD) from 11 imaging centers who underwent serial CCTA examinations were retrospectively analyzed. Plaque volumes and PCAT were quantified, and the presence of high-risk plaque features was semi-quantitatively assessed using the *plaque feature score* (PFS). **Results**: In total, 216 consecutive patients (mean age 63.1 ± 9.7 years, 26.4% female) were included. The mean observation and treatment timespan between the CCTA scans was 824.5 (interquartile range (IQR) = 463.0–1323.0) days (27.5 months). The regression of high-risk features was more common with high-intensity versus low or no lipid-lowering treatment (HR = 4.6, 95%CI = 1.8–12.0, *p* < 0.001) and was associated with the attenuated increase in non-calcified plaque volume (*p* < 0.001). PCAT_mean_ decreased with increasing intensity of lipid-lowering treatment (*p* = 0.01) but no associations were observed between the changes in PCAT and PFS or plaque volumes. Lipid-lowering drug intensity was predictive of PFS regression (*p* < 0.001), whereas baseline PCAT_RCA_ was predictive for PFS progression (*p* = 0.03), both independent of age, cardiovascular risk factors, and baseline plaque volumes. **Conclusions**: PCAT predicts the progression of high-risk coronary plaque features. High-intensity lipid-lowering drugs may cause the regression of high-risk plaque features through a plaque ‘delipidization’ process. Future trials are now warranted, studying if this process is potentially associated with improved clinical outcomes.

## 1. Introduction

Due to technical advances and recent landmark clinical randomized controlled trials (RCTs) [1,2], coronary computed tomography angiography (CCTA) has emerged as the first-line diagnostic tool for the non-invasive work-up and risk stratification of patients with coronary artery disease (CAD) [3,4,5,6]. Although serial CCTA examinations are not generally recommended by current guidelines, previous observational trials and meta-analyses highlighted the possible role of CCTA in evaluating the effect of lipid-lowering treatment on plaque volumes and composition [7,8]. Previous studies such as the PARADIGM trial investigated such effects by treating statin treatment in a binary manner [9]. Therefore, data on the influence of lipid-lowering treatment differentiating between moderate- and high-intensity lipid-lowering regimes on preexisting high-risk plaque features are scarce [10]. In addition, pericoronary adipose tissue (PCAT) attenuation has been described as a prognostic marker, capturing inflammatory risk beyond conventional clinical and CCTA parameters [11], but its role on plaque progression and PCAT changes during lipid-lowering treatment have not been investigated so far.

In the LOCATE study, we recently demonstrated the ability of moderate- and high-intensity statin treatment and PCSK9 inhibition to attenuate and arrest the progression of non-calcified coronary plaque atheroma, respectively, [12]. In the present retrospective study, we sought to investigate the underlying mechanisms behind plaque changes by examining the impact of drug intensity on high-risk plaque features and associations between plaque features and plaque volume changes. In addition, we sought to investigate if different intensities of lipid-lowering treatments can affect PCAT during serial CCTA scans and if baseline PCAT is associated with the progression of high-risk plaque features.

## 2. Methods

### 2.1. Patient Population

The present study includes individuals with suspected or known CAD from 11 cardiology- or radiology-based centers who underwent serial CCTA examinations at intervals of at least 6 months. All participants presented with stable clinical symptoms, whereas patients undergoing CCTA in the setting of acute coronary syndromes (ACSs) or for alternative indications, including pre-procedural planning of structural cardiac interventions, were excluded from the analysis. Details on study patient enrollment, study protocol and scanner model/vendor, detector configuration, slice count/detector rows, and reconstructive software are provided in Appendix A and are also reported elsewhere [12].

Cardiovascular (CV) risk factors, including all traditional atherogenic risk factors and history of myocardial infarction or prior percutaneous coronary intervention (PCI) were recorded. The intensity of lipid-lowering treatment was classified as low, moderate, or high intensity based on the type of medication and dose according to current recommendations [9]. Thus, 5 mg of atorvastatin, 10 mg of simvastatin, 10–20 mg of pravastatin, and 20–40 mg of fluvastatin were deemed as low intensity; 10–20 mg of atorvastatin, 5–10 mg of rosuvastatin, 20–40 mg of simvastatin, 40–80 mg of pravastatin, and >40 mg of fluvastatin were classified as intermediate intensity; whereas 40–80 mg of atorvastatin, 20–40 mg of rosuvastatin, or PCSK9 inhibitors were classified as high-intensity treatments. Notably, the decision for the initiation of low-, medium-, or high-intensity treatment was not part of the study protocol since this decision was left at the discretion of the imaging and clinical physicians. In addition, since the documentation of medications started after the baseline CCTA scan, we refer to the lipid-lowering medication prescribed after the baseline CCTA as “baseline medication”.

Ethical approval and a waiver of written informed consent for all patients were granted by the local ethics committee (S-925/2021, approved date: 26 December 2021) of the University of Heidelberg (Alte Glockengießerei 11/1, 69115 Heidelberg, Germany), as described elsewhere [12]. No confidential patient information was collected or exchanged. Additional ethical and regulatory approvals were obtained locally at each individual institution based on local regulations. The study complied with the Declaration of Helsinki. The local principal investigator from each institution was responsible for data collection, transfer, and local approval. A fully anonymized, uniform, electronic reporting form was maintained across all institutions. The study was registered on the German Clinical Trials Register (DRKS00031954, 06/2023).

### 2.2. CCTA Examinations

The CCTAs were executed by using at least 64-slice CT scanners. Appendix A presents the number of examinations performed with each scanner at baseline and follow-up, together with an overview of scanner-specific parameters.

Patients with a heart rate of >65 bpm were given metoprolol intravenously in individually adjusted doses to attain a heart rate of <65 bpm. Using a tube voltage of either 100 or 120 kV, native coronary calcium scans were acquired prior to the administration of the contrast agent. The acquisition protocol was selected individually based on patient-specific parameters, including heart rate, rhythm, and coronary calcification.

### 2.3. CCTA Analysis

CCTA scan series were evaluated independently by seasoned readers, each possessing more than five years of experience in CCTA and certified by the German Society of Cardiology (GK, SG, and MS). This certification corresponds to level 3 of clinical competence training as defined by the Society of Cardiovascular Computed Tomography [13].

### 2.4. Analysis of Agatston Score, CAD-RADS 2.0 and Plaque Volume Quantification

Coronary calcium scans were utilized to quantify the Agatston score, representing the total coronary calcium score. The degree of luminal stenosis was assessed according to the CAD-RADS 2.0 (coronary artery disease reporting and data system) classification: 0, none (0%); 1, minimal (1–24%); 2, mild (25–49%); 3, moderate (50–69%); 4, severe (70–99%); and 5, chronic total occlusion (100%) [14].

Structures within or adjacent to the coronary lumen exceeding 1 mm^2^ were defined as a coronary plaque. Quantification of plaque volume was measured using the CT Coronary Plaque Analysis prototype (version V30, Siemens Healthineers, Forchheim, Germany), a new semi-automated prototype plaque evaluation software. The software employs deep-learning algorithms to automatically delineate coronary centerlines, lumen, and vessel wall contours. By applying Hounsfield Unit (HU)-based thresholding (>350 HU: calcified plaque; <350 HU: non-calcified plaque, including fibrous (30–350 HU) and lipid (<30 HU) plaque components), the software employs deep-learning algorithms to automatically delineate coronary centerlines, lumen, and vessel wall contours and automatically determine plaque volumes for the complete coronary tree [14]. This new prototype software tool has provided excellent reproducibility for the assessment of different plaque components (inter- and intra-observer variabilities of less than 5% using the CT-guided PCI software analysis tool) [15,16]. Plaque burden was measured in coronary arteries of patients who exhibited at least one lesion with luminal narrowing >25% in one or more coronary artery segments.

In each patient, baseline and follow-up CCTA scans were used to quantify total, calcified, and non-calcified plaque volumes. Changes in volumes between the baseline and follow-up CCTA were interpreted as plaque progression when increasing, or regression when decreasing. Reduction in non-calcified plaque was referred to as plaque ‘delipidization’.

### 2.5. Analysis of High-Risk Plaque Features

The presence of high-risk plaque features, including (i) low-attenuation plaque burden (voxel < 30 HU within a coronary plaque), (ii) positive remodeling (remodeling index > 1.1), (iii) spotty calcification (calcified burden < 3 mm within a non-calcified plaque), or (iv) the napkin-ring sign (low-attenuation core (HU ≤ 70) with circumferential high attenuation ring) were evaluated in each coronary vessel [16,17]. Subsequently, a score was generated based on (i) the absence of plaques (0 points), (ii) the presence of plaque without high-risk features (factor = 1 point), (iii) the presence of plaque with a single high-risk feature (factor = 2 points), and (iv) the presence of plaque of two or more high-risk features (factor = 3 points). Based on this semiquantitative analysis, a *plaque feature score* (PFS), which correlates with the amount of plaque features, was generated per patient and scanned by adding all plaques of the coronary tree after multiplying each plaque with the corresponding factor:PFS baseline = plaque#1*factor(1–3) + plaque#2*factor(1–3) + plaque#3*factor(1–3) + plaque#n*factor(1–3)

The DPFS was calculated in every patient by considering the difference in plaques with high-risk plaque features between the baseline and follow-up CCTA and shows progression (DPFS > 0) or regression of high-risk features (DPFS < 0) on serial observation.DPFS = PFS follow-up CCTA − PFS baseline CCTA

The assessment of plaque volumes and high-risk plaque features was performed, blinded to the clinical data of the patients and to the intensity of lipid-lowering treatments.

### 2.6. PCAT Analysis

Pericoronary adipose tissue (PCAT) was quantified using the research plaque evaluation software (syngo.via Frontier CT Coronary Plaque Analysis, version 5.0.2, Siemens Healthineers, Forchheim Germany) and expressed in Hounsfield Units (HU) [18]. For the measurement of PCAT, the proximal 4 cm segments of all three coronary vessels (right coronary artery (RCA), left anterior descending artery (LAD), and left circumflex artery (LCX)) were delineated. Perivascular fat was defined as adipose tissue located within a radial distance from the outer vessel wall equivalent to the vessel diameter [11,16,18]. To adjust for attenuation between scans performed at different tube voltages, the PCAT attenuation values obtained at tube voltages below 120 kVp were divided by their corresponding conversion factors, enabling comparability with scans performed at 120 kVp (e.g., conversion factor of 1.114 for scans performed at 100 kVp and of 1.267 for scans performed at 80 kVp, as previously reported [16]. In addition, the PCAT_mean_ was calculated as a mean of the PCAT of the RCA, LAD, and LCX. PCAT was available only in a subset of patients. In 70 (32.4%) subjects, PCAT could not be assessed due to limited compatibility of the corresponding CCTA-files from different vendors with the available analysis software.

### 2.7. Statistical Analysis

Statistical analyses were performed using the MedCalc 20.009 software (MedCalc software, Mariakerke, Belgium). Continuous normally distributed variables are expressed as mean ± standard deviation, whereas non-normally distributed variables are reported as median and interquartile range (IQR). Normal distribution was tested using the Shapiro–Wilk test. Categorical variables are reported as numbers and percentages. Categorical data were compared using chi^2^ tests. For correlation analysis, Pearson correlation coefficients were calculated and reported, including 95% confidence intervals (CIs). The ANOVA test was used for comparing three or more normally distributed groups with the Scheffé test for post hoc analysis, whereas for continuous variables, which were not normally distributed, non-parametric tests (Mann–Whitney U-tests or Kruskal–Wallis tests) were performed. Kaplan–Maier curves were used to evaluate PFS changes in patients with different lipid-lowering treatment regimens. Cox-proportional hazard regression models considering the timespan between baseline and follow-up CCTA were used to assess the value of PCAT_mean_ for the prediction of changes in plaque features after adjustment for age, CV risk factors, baseline plaque volume, and lipid-lowering treatment intensity. Kappa statistics were used for the calculation of inter- and intra-observer agreement for the assessment of high-risk plaque features in 120 randomly selected coronary vessels. In addition, propensity matching was performed after adjusting clinical and imaging characteristics such as age (maximum allowable difference of 5 years), total number of cardiovascular risk factors (exact match required), and baseline plaque volumes by quartiles (exact match required) in patients with no/low- versus moderate-/high-intensity treatment, resulting in 40 matched cases. Furthermore, a receiver operating characteristic (ROC) analysis was performed to assess the ability of the treatment intensity to predict the attenuated progression of non-calcified atherosclerotic plaque using all timepoints during follow-up as well as two randomly selected time-interval subgroups. Differences were considered statistically significant at *p* < 0.05.

## 3. Results

### 3.1. Demographic Data

In total, 216 consecutive patients (63.1 ± 9.7 years old, 57 (26.4%) female) underwent serial CCTAs (median timespan of 824.5 (IQR = 463.0–1323.0) days, (27.5 months)). Demographic data are provided in Table 1. The mean number of antihypertensive comedications at baseline was 1.0 (0–2.0) per patient, whereas 95 (44.0%) patients where on antiplatelet agents. After the baseline CCTA scans, 89 (41.2%) patients received no or low lipid-lowering treatment, 80 (37.0%) were initiated on moderate-intensity, and 47 (21.8%) on high-intensity lipid-lowering treatment. Details on lipid drugs are provided elsewhere [12]. Serial CCTA scans were performed using the same vendor in all cases (100%) and the same scanner type in 182 (84.3%) cases.

### 3.2. High-Risk Plaque Features and Effects of Lipid-Lowering Treatment

During baseline and follow-up CCTA, 68 (31.5%), and 59 (27.3%) patients showed 84 and 73 plaques, respectively, with at least one high-risk feature. From 157 plaques with 175 high-risk features in total (including baseline and follow-up CCTA scans), low-attenuation plaque was the most common high-risk feature detected in 100 (57.1%) cases, followed by positive remodeling, spotty calcification, and the napkin-ring sign in 39 (22.3%), 25 (14.3%), and 11 (6.3%) cases, respectively, (Table 2). During follow-up CCTA, 29 plaques with new-onset high-risk features were detected in 26 (12.0%) patients, whereas high-risk features regressed in 40 (18.5%) cases.

Progression versus regression of high-risk plaque features was noted in patients with no or low- versus moderate- and high-intensity treatment (chi^2^ = 17.2, *p* = 0.03) (Figure 1A). The regression of high-risk features over time was significantly more frequent with high- versus moderate- and no or low-intensity lipid-lowering treatment (HR = 4.6, 95%CI = 1.8–12.0, *p* < 0.001) (Figure 1B). After propensity matching, a non-significant trend remained for high-risk plaque feature regression during high/moderate versus no/low-intensity treatment (Appendix A). In addition, attenuated plaque progression was predicted by intensity of the treatment within randomly selected time-interval subgroups (Appendix A).

### 3.3. Changes in Plaque Volumes and PCAT

Significant changes in the total and non-calcified plaque volumes were noted dependent on the intensity of the lipid-lowering drugs (Figure 2A,B). No association was noted; however, between the calcified plaque volume and the Agatston score (*p* = NS).

In addition, significant absolute and relative PCAT_mean_ changes (decrease in absolute PCAT_mean_ values in C and relative increase in PCAT_mean_ values in D) were observed with increasing intensity of the lipid-lowering treatment (Figure 2C,D).

### 3.4. Association of Plaque Features with Plaque Volumes, Agatston Score, and PCAT

The progression and regression of high-risk plaque features were related to changes in total and non-calcified volumes (Figure 3A,B, *p* < 0.001 for both) but not with changes in calcified plaque volume and Agatston score (*p* = NS for both).

A significant association was found between reduction in non-calcified plaque volumes and decreasing LDL-cholesterol values (r = 0.33, 95%CI = 0.14–0.55, *p* = 0.001). No association could be found, however, between the progression or regression of high-risk plaque features or plaque volumes with changes in PCAT_RCA_, PCAT_LAD_, PCAT_LCX_, and PCAT_mean_ (*p* = NS for all). In addition, no associations were present between changes in PCAT and changes in plaque volumes or DPFS (Appendix A) and between LDL-level and PCAT changes (r = 0.006, *p* = 0.95).

### 3.5. Prediction of High-Risk Plaque Feature Progression and Regression

Lipid-lowering treatment intensity and baseline total plaque volume predicted high-risk plaque regression, independent of age and CV risk factors (Table 3A).

Progression of high-risk plaque features was associated with baseline PCAT_RCA_, independent of lipid-lowering treatment intensity, baseline total plaque volumes, age, and CV risk factors (Table 3B).

The number of antihypertensive drugs, but not antiplatelet treatment at baseline, was predictive of high-risk plaque feature progression by univariate analysis (Appendix A). The number of antihypertensive drugs, however, was not independently predictive for plaque progression or regression (Appendix A).

### 3.6. Observer Variabilities

Intra- and inter-observer agreement for the assessment of high-risk plaque features was k = 0.71 (95%CI = 0.57–0.84) k = 0.61 (95%CI = 0.45–0.77), respectively.

### 3.7. Case Examples

Representative examples of high-risk plaque features as well as plaque volumes and features regression versus progression during high- versus no/low-intensity lipid-lowering treatment can be appreciated in Figure 4A–D.

## 4. Discussion

The results of the present multi-center observational study demonstrate that high-intensity lipid-lowering statins and PCSK9 inhibitors are strongly associated with regression of high-risk plaque features, which have been described as precursors of ACS [19,20]. In addition, the regression of plaque features seems to be part of a plaque ‘delipidization’ process since the retreat of such high-risk features is strongly related to reductions in non-calcified plaque volumes. Furthermore, changes in PCAT are observed during treatment with statins, which are not related to changes in plaque features or volumes. However, baseline PCAT_RCA_ seems to be an important imaging biomarker, capturing pro-inflammatory activity since PCAT_RCA_ is related to progression of high-risk plaque features independent of conventional risk factors and of statin treatment intensity (Appendix A). Finally, LOCATE emphasizes once again that CCTA is a valuable tool for the serial evaluation of high-risk plaque features and for the monitoring of the mid-term effects of lipid-lowering treatment drug intensities.

ACC/AHA guidelines have endorsed the use of high-intensity lipid-lowering treatments in patients at high risks of future cardiac events [9]. Likewise, European Society of Cardiology (ESC) guidelines recommend aggressive lipid lowering, aiming at >50% LDL-C reduction from baseline and <55 mg/dL in patients with CAD, peripheral artery, or carotid disease [21]. Recently, regression of atherosclerotic plaque volume by 1% was shown to dramatically improve MACE rates by 25% in patients with atherosclerotic disease [22]. In addition, observational studies and metanalyses, as well as the LOCATE study, recently confirmed a link between high-intensity statin treatment and coronary plaque volume regression, thus corroborating existing guideline recommendations that endorse high-intensity lipid-lowering interventions for patients at high cardiovascular risk [8,12].

### 4.1. Previous Studies on Plaque Features and Prognosis

Several studies have already demonstrated that high-risk plaque features can serve as potential precursors to plaque rupture, leading to cardiac events [20,23,24], A secondary analysis of the PROMISE study, in which the authors attempted to determine the association between high-risk plaque features and major adverse cardiovascular events (MACEs), showed that the presence of such features was associated with a higher risk for future MACEs (6.4% vs. 2.4%) [25]. In addition, the ROMICAT II study revealed that the presence of high-risk plaque features was significantly associated with ACS in 472 patients who underwent coronary CT angiography (CCTA) for diagnostic evaluation due to acute chest pain [26]. In the same direction, Taron et al. showed that high-risk plaque features provided superior prognostic value compared with total plaque burden for the prediction of clinical endpoint, like unstable angina, myocardial infarction, and death [23]. In addition, high-risk plaques with low attenuation proved to be better predictors of cardiac events than lumen narrowing [5]. A meta-analysis of six studies emphasized that plaques with high-risk features are significantly more likely to result in future ACS, underscoring the importance of this parameter for the robust risk stratification in patients with chronic coronary syndrome (CCS) [27].

The role of statin treatment on high-risk plaque features during lipid-lowering therapy has been investigated in the PARADIGM trial [28,29]. This trial showed that statins were associated with a 35% reduction in high-risk plaque feature development [28]. This was also confirmed in a study by Park et al., where statin treatment caused attenuated plaque progression, especially in patients with a higher number of high-risk plaque features [10]. Conversely, high-risk plaque features have been described as a precursor of rapid plaque progression, potentially causing acute cardiac events and death. However, statin treatment was rated categorically in all these previous trials, without differentiating between low-, moderate-, and high-intensity drugs, which distinguishes the present study from previous reports.

Thus, in LOCATE, changes in high-risk plaque features were observed during different intensities of lipid-lowering drugs. Especially with high-intensity treatment, four-fold higher rates of plaque regression were noticed compared with no or low-intensity lipid-lowering treatment. The retreat of such high-risk features over time can be regarded as a reversed remodeling process, which is strongly related to lipid-lowering treatment intensity and is accompanied by non-calcified volume reductions as a part of a plaque ‘delipidization’ process with concomitant reduction in pericoronary tissue inflammatory activity, as measured by the PCTA attenuation. Notably, CCTA image quality needs to be considered when comparing serial images from different scanners and vendors regarding PFS and DPFS since different image resolution with different vendors may substantially influence these parameters, including the analysis of PCAT. In our study, however, the same vendors and in most cases the same scanner types were used with the initial and follow-up CCTA scans. In addition, the agreement between readers for the evaluation of high-risk plaque features was rather moderate, which is confirmatory to recent observations and points to the need for quality-controlled education programs for advanced cardiac imaging, which is necessary to achieve standardization in the assessment of such complex metrics [30]. This is also important from a clinical perspective since the detection of such plaques may trigger the prescription of high-intensity lipid-lowering or/and antiplatelet agents.

### 4.2. The Role of PCAT, Association with Plaque Burden and Influence of Statin Treatment

PCAT encases the epicardial coronary artery and can directly interact with the vessel wall through paracrine signaling in a bidirectional manner [31]. Recent studies suggested that PCAT was linked to atherosclerosis progression, as a marker of coronary inflammation [32], and was also a prognostic marker for CV events [18]. In addition, an association between PCAT and high-risk plaque features has been reported, where increasing PCAT was associated with increased plaque volumes and higher numbers of plaques with high-risk features [16]. Furthermore, high-risk plaque features and PCAT showed complementary values for the prediction of cardiac events in patients with CCS [33,34]. The role of PCAT was reinforced by a meta-analysis conducted in 6335 patients, where less negative PCAT values were associated with a markedly higher risks of MACEs during mid- and long-term follow-up with a hazard ratio of 3.29 [35]. The recently published ORFAN study demonstrated that PCAT captures inflammatory risks beyond current clinical risk management and CCTA parameters, especially in patients with non-obstructive CAD [11]. Patients with mild CAD and less negative PCAT values in all three coronary arteries exhibited an increased number of cardiac deaths and other MACEs during follow-up [11]. However, the role of PCAT was recently questioned in another recent study, where PCAT values failed to predict MACEs during long-term follow-up of 9.5 years [36]. Therefore, the prognostic utility of PCAT requires further validation in future studies. In our study, PCAT predicted plaque progression, which supports an inflammatory phenotype signal but was outperformed by other variables, like baseline plaque volume, for the prediction of high-risk plaque feature regression, which needs to be considered in future studies. In addition, the absence of a direct correlation between alterations in PCAT and plaque regression may imply that anti-inflammatory effects of lipid-lowering therapy may be facilitated by mechanisms not entirely represented by PCAT metrics, highlighting the intricacy of vascular biology or may be related to limited power, measurement errors, or asynchronous effects of lipid versus inflammatory pathways. Notably, PCAT around the RCA was shown to be a representative biomarker of global coronary inflammation, predicting adverse outcomes in the previous landmark study by Oikonomou et al. [18]. However, in the more recent ORFAN study [11], increased PCAT in all the three coronary arteries had an additive impact on the risk for cardiac mortality. In our study, we therefore calculated the mean of the PCAT in all the three coronary arteries, considering both PCAT_mean_ and PCAT_RCA_ in our main analysis based on the two above-mentioned large-scale studies [11,18]. Interestingly, the number of antihypertensive agents predicted PFS progression over time. This may indicate that patients with more challenging blood pressure control may be at a higher risk for high-risk plaque feature progression over time. This effect, however, was not statistically significant after the consideration of other parameters like baseline PCAT_RCA_, which in this respect remained significant for PFS progression.

In addition, although PCAT is in the meantime a recognized imaging biomarker based on the above-mentioned scientific evidence [11,16,18,31,33,35], its clinical use during CCTA examinations remains limited. In this regard, previous studies were limited to stable patients and did not evaluate the natural history of PCAT attenuation or the impact of statin therapy on PCAT, which is awaited in the ongoing ARISTOCRAT study [37]. Notably, a smaller study including 170 patients with type 2 diabetes mellitus, treated with evolocumab versus standard lipid-lowering therapy, reported a decrease in PCAT density of the right coronary artery in the evolocumab group, which was not related to changes in LDL, but rather to changes in lipoprotein (a) levels [38]. The changes observed by Yu et al. are in accordance with the present results, since the decrease in PCAT attenuation was primarily observed in the high-intensity lipid-lowering treatment group, where ~50% of the patients received PCSK9 inhibition. However, the effects of lipid-lowering treatment on PCAT were small, and the clinical significance of these findings merited further investigation in future trials, where PCAT attenuation together with changes in plaque burden and changes in biomarkers of cardiac and vascular injury [6,20,39] and inflammation needed to be analyzed prospectively.

### 4.3. Limitations

Our study has some limitations. First, the retrospective nature of our register needs to be acknowledged, which did not allow for the control for treatment adherence to statins and of possible treatment interruptions, thus weakening causal inference between therapy intensity and plaque changes over time [40]. In the same direction, only patients undergoing repeat CCTA were included, potentially excluding high-risk or non-adherent patients. In addition, the duration of the follow-up period varied between individual patients and centers (Appendix A), whereas lipid-lowering therapy prior to the index CCTA examinations (‘pre-baseline’) were not recorded, which is a limitation. Furthermore, although serial CCTA scans were performed with the same vendors in all cases, different scanner types were involved in some cases, which may have influenced the quantification of plaque volumes and PCAT. PCAT values were available only in ~70% of the patients and in some cases different tube voltages have been used between baseline and follow-up scans, which both are limitations. However, the missing PCAT data were attributed to incompatibility of the corresponding datasets, which does not necessarily introduce additional selection bias since CCTA-files incompatibility is not expected to be related to clinical or imaging characteristics, such as lower patient compliance, more severe disease, different socioeconomic status, etc. In this regard, key clinical and imaging characteristics were not significantly different between the patients with versus without available PCAT data (Appendix A). In addition, conversion factors were used for adjustment of the PCAT values, as previously described [16]. In the same direction, changes with PCAT attenuation were of smaller extent compared with those observed with plaque volumes and high-risk plaque features, which may be associated with a less pronounced impact of statins on PCAT attenuation. In this regard, future more specific anti-inflammatory therapies may be necessary to modify this imaging biomarker, which may deem as a therapeutic target in future trials [37]. In addition, comedications, like antidiabetics and their effects on risk factor control during the study period, and changes in BMI over time, possibly contributing to plaque progression or regression over time, were not controlled by our retrospective study design, which is a limitation. Furthermore, evaluating the impact of plaque changes over time on cardiac outcomes was beyond the scope of the present observational trial. In this regard, prospective long-term multi-center trials are now warranted, incorporating standardized imaging intervals, stringent adherence monitoring, and biochemical markers of inflammation in conjunction with imaging endpoints, to investigate the role of PCAT and lipid-lowering treatment on plaque progression and cardiac outcomes.

## 5. Conclusions

The multi-center LOCATE study underscores the role (i) of PCAT, as a surrogate marker of pericoronary inflammation, mediating the progression of high-risk plaque features, and (ii) of lipid-lowering treatment, mediating atherosclerosis regression. High-intensity lipid-lowering treatment with potent statins and PCSK9 inhibitors is associated with plaque ‘delipidization’ and with the regression of high-risk plaque features, as a part of a plaque stabilization process. Notably, the link to a prognostic benefit cannot be provided within our study but such changes in plaque composition were previously shown to be associated with substantial reduction in MACEs [22]. Changes in PCAT are also observed during treatment with statins but are not directly related to changes in plaque features or volumes, which indicates additional mechanisms of action.

## Figures and Tables

**Figure 1 diagnostics-15-02340-f001:**
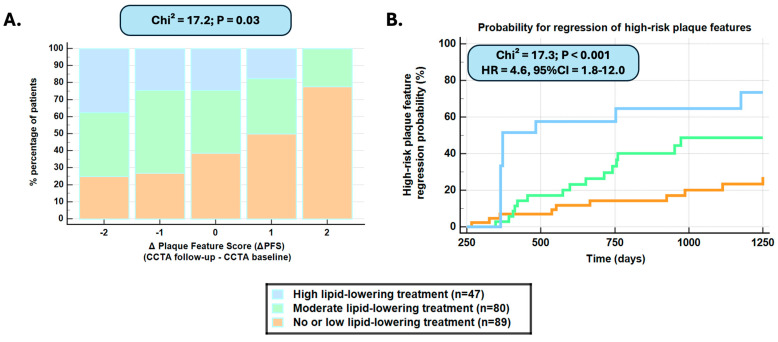
Regression of high-risk plaque features based on the plaque feature score (ΔPFS) was observed in patients with moderate and high versus low/no treatment (*p* = 0.03) (**A**). Regression of high-risk plaque features was observed significantly more frequently with high- followed by moderate- versus low-/no intensity lipid-lowering treatment (HR = 4.6, 95%CI = 1.8–12.0, *p* < 0.001 for high- versus low-intensity or no lipid-lowering treatment) (**B**).

**Figure 2 diagnostics-15-02340-f002:**
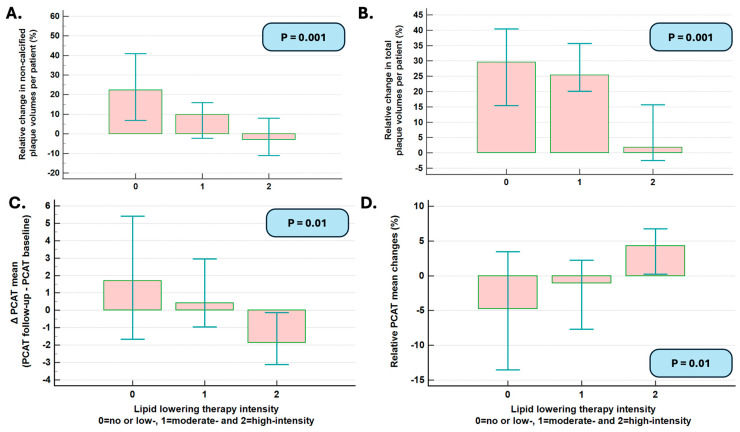
Attenuated progression of non-calcified (**A**) and total plaque volume (**B**) was noted in patients with moderate versus no/low-intensity treatment, whereas plaque progression stopped during high-intensity lipid-lowering treatment. PCAT_mean_ values decreased with increasing intensities of lipid-lowering treatments (**C**,**D**).

**Figure 3 diagnostics-15-02340-f003:**
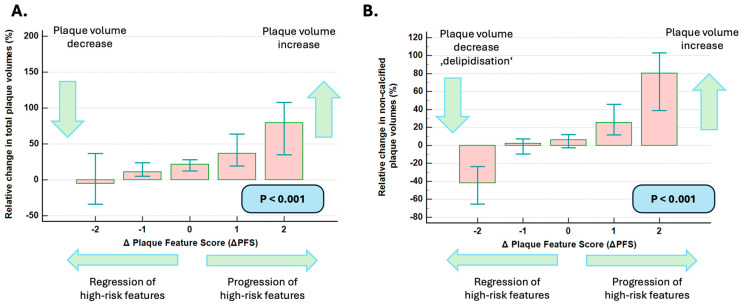
Progression and regression of high-risk plaque features, respectively, were related to changes in total and non-calcified plaque volumes (**A**,**B**). Retreat of high-risk plaque features was thereby linked to a ‘delipidization’ process accompanied by reductions in total (**A**) and non-calcified (**B**) plaque volumes.

**Figure 4 diagnostics-15-02340-f004:**
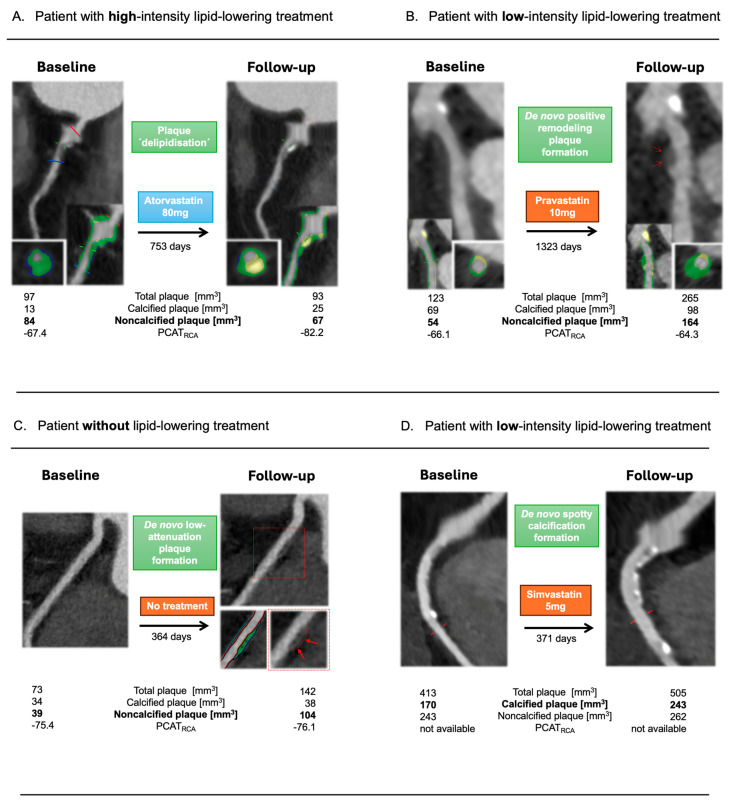
(**A**–**D**). In patient **A**, treated with 80 mg of atorvastatin over 2.1 years, non-calcified plaque decreased, whereas calcified components nearly doubled, thus resulting in plaque ‘delipidization’. In patient **B**, treated with 10 mg of pravastatin over 3.6 years, an increase in the non-calcified plaque volume was noted as well as a de novo positive remodeled plaque, which was not present during the baseline CCTA scan. Patient **C** showed, within one year without any lipid-lowering treatment, an increase in the non-calcified plaque volume, resulting in a de novo low-attenuation plaque, while in patient **D**, a de novo spotty calcification was noted during the follow-up CCTA scan.

**Table 1 diagnostics-15-02340-t001:** Clinical, laboratory, and baseline CCTA data, by lipid-lowering treatment intensity.

	No or Low-Intensity Therapy *n* = 89	Moderate-Intensity Therapy *n* = 80	High-Intensity Therapy *n* = 47	*p*-Values
	**Baseline data and risk factors**			
Age (yrs.)	60.7 ± 10.9	66.4 ± 8.7	62.2 ± 7.5	<0.001
Female sex	26 (29.2%)	19 (23.8%)	12 (25.5%)	0.71
Body-mass index (kg/m^2^)	26.6 (24.1–28.9)	27.8 (25.5–31.0)	29.4 (26.3–34.0)	0.01
Arterial hypertension *	45 (51.7%)	68 (87.2%)	28 (80.0%)	<0.001
Hyperlipidemia **	48 (55.2%)	66 (83.5%)	39 (95.1%)	<0.001
Diabetes mellitus ***	5 (5.7%)	14 (17.7%)	7 (20.0%)	0.03
Active or former smoking §	25 (32.1%)	16 (21.6%)	13 (35.1%)	0.22
Family history of CAD §§	24 (32.0%)	22 (34.9%)	21 (61.8%)	0.009
Total number of CV risk factors	1.0 (1.0–2.0)	2.0 (2.0–3.0)	2.0 (2.0–3.0)	<0.001
	**History of CAD**			
History of CAD	11 (12.4%)	31 (38.8%)	26 (55.3%)	<0.001
Prior PCI	10 (11.2%)	23 (28.8%)	22 (46.8%)	<0.001
Prior myocardial infarction	6 (6.7%)	6 (7.5%)	11 (23.4%)	0.006
	**Clinical presentation at baseline**			
Stable chest pain syndrome	62 (69.7%)	31 (38.8%)	11 (23.4%)	<0.001
Exertional dyspnea	45 (50.6%)	28 (35.0%)	11 (23.4%)	0.005
Palpitations/unspecific symptoms	17 (19.1%)	27 (33.8%)	16 (34.0%)	0.004
Syncope	2 (2.3%)	0 (0.0%)	1 (2.1%)	0.55
	**Baseline medications (as prescribed after the baseline CCTA scans)**			
Aspirin	23 (25.8%)	42 (52.5%)	26 (55.3%)	<0.001
Aspirin or P2Y12 inhibitors	25 (28.1%)	43 (53.8%)	27 (57.4%)	<0.001
ß-blockers	29 (32.6%)	40 (50.0%)	27 (57.5%)	<0.01
Calcium antagonists	9 (10.1%)	21 (26.3%)	9 (19.2%)	0.01
Diuretics	9 (10.1%)	24 (30.0%)	7 (14.9%)	0.003
ACE inhibitors or AT2 blockers	26 (29.2%)	43 (53.8%)	12 (25.5%)	<0.001
Number of antihypertensive medications	0 (0–1.0)	2.0 (1.0–2.0)	1.0 (0–2.0)	<0.001
PCSK9 inhibitors	0 (0.0%)	0 (0.0%)	24 (51.1%)	<0.001
Statins	9 (10.1%)	80 (100.0%)	38 (80.9%)	<0.001
Ezetimibe	2 (2.3%)	9 (11.3%)	13 (27.7%)	<0.001
	**Baseline laboratory data**			
Hemoglobin(mg/dL)	14.5 (13.5–15.2)	14.4 (13.8–15.4)	14.6 (13.2–15.4)	0.87
Estimated GFR (ml/min/1.73 m^2^)	84.6 (69.1–94.2)	83.2 (72.6–92.5)	87.6 (73.0–94.0)	0.84
Creatinine (mg/dL)	0.95 (0.81–1.06)	0.93 (0.80–1.05)	0.90 (0.82–1.05)	0.90
Total cholesterol (mg/dL) #	213.0 (175.0–244.5)	181.0 (149.3–214.5)	187.5 (150.5–222.5)	0.03
LDL-cholesterol (mg/dL) ##	127.0 (96.3–159.5)	94.0 (76.5–126.3)	121.0 (80.3–154.5)	0.02
	**Baseline CCTA parameters**			
Agatston score	42.3 (2.5–202.0)	139.7 (46.3–348.0)	286.3 (108.0–824.6)	0.007
CAD RADS 2.0	1.0 (1.0–1.0)	1.0 (1.0–2.0)	2.0 (1.0–2.0)	<0.001
Total plaque volume (mm^3^)	208.0 (60.0–398.0)	399 (191.5–923.5)	867.0 (400.8–1543.5)	<0.001
Non-calcified plaque volume (mm^3^)	165 (55.8–309.0)	278.5 (116.5–493.5)	378.0 (194.5–599.3)	<0.001
Calcified plaque volume (mm^3^)	32 (4.8–103.8)	120.0 (48.5–362.5)	319.0(190.0–784.0)	<0.001
Plaque feature score (PFS)	2.0 (2.0–3.0)	2.5 (2.0–3.0)	3.0 (2.0–4.0)	<0.001
PCAT RCA †	−70.4 (−79.3 to −66.4)	−68.9 (−77.5 to −62.3)	−69.8 (−74.1 to −66.0)	0.58
PCAT LAD †	−74.2 (−77.8 to −68.0)	−70.9 (−75.0 to −65.1)	−71.8 (−75.1 to −68.2)	0.04
PCAT LCX †	−68.6 (−73.1 to −63.4)	−65.8 (−69.9 to −61.3)	−66.4 (−69.7 to −60.2)	0.05
PCAT Mean †	−70.3 (−76.8 to −65.8)	−68.1 (−74.2 to −63.9)	−68.9 (−72.5 to −66.0)	0.12

PCI, percutaneous coronary intervention; GFR, glomerular filtration rate; AT, angiotensin; ACE, angiotensin converting enzyme; CAD, coronary artery disease; LDL, low-density lipoprotein; CCTA, coronary computed tomography angiography; CV, cardiovascular; PCSK9, proprotein convertase Subtilisin/Kexin type 9. * Values not available in 16 (7.4%) patients; ** values not available in 9 (4.2%) patients; *** values not available in 15 (6.9%) patients; § values not available in 27 (12.5%) patients; §§ values not available in 44 (20.4%) patients; # values available in 122 (56.5%) patients; ## values available in 114 (52.8%) patients; † values available in 147 (68.1%) patients.

**Table 2 diagnostics-15-02340-t002:** Number and type of high-risk plaque features by coronary artery territory and timepoint.

	Low Attenuation		Positive Remodeling		Spotty Calcification		Napkin-Ring Sign		Total Number of High-Risk Features	
	Baseline CCTA	Follow-Up CCTA	Baseline CCTA	Follow-Up CCTA	Baseline CCTA	Follow-Up CCTA	Baseline CCTA	Follow-Up CCTA	Baseline CCTA	Follow-Up CCTA
**RCA**	17 (18.1%)	13 (16.0%)	6 (6.4%)	5 (6.2%)	4 (4.3%)	2 (2.5%)	5 (5.3%)	3 (3.7%)	32 (34.0%)	23 (28.4%)
**LAD**	25 (26.6%)	21 (25.9%)	10 (10.6%)	11 (13.6%)	8 (8.5%)	7 (8.6%)	1 (1.1%)	1 (1.2%)	44 (46.8%)	40 (49.4%)
**LCX**	13 (13.8%)	11 (13.6%)	3 (3.2%)	4 (4.9%)	2 (2.1%)	2 (2.5%)	0 (0.0%)	1 (1.2%)	18 (19.1%)	18 (22.2%)
**Total per scan**	55 (58.5%)	45 (55.5%)	19 (20.2%)	20 (24.7%)	14 (14.9%)	11 (13.6%)	6 (6.4%)	5 (6.2%)	**94 (53.7%)**	**81 (46.3%)**
**Total**	**100 (57.1%)**		**39 (22.3%)**		**25 (14.3%)**		**11 (6.3%)**		**175 (100.0%)**	

RCA indicates right coronary artery; LCX, left circumflex artery; and LAD, left anterior descending artery. Total per patient including baseline and follow-up CCTA scans.

**Table 3 diagnostics-15-02340-t003:** **A**–**B.** Cox-proportional hazard regression models for the prediction of high-risk plaque feature regression (**A**) and progression (**B**).

**A. Regression of high-risk plaque features**	Coefficient	Standard error	Wald	Hazard ratio	95% Cl	*p*-values
Age	0.001	0.022	0.006	1.001	0.95 to 1.04	0.93
Total number of CV risk factors	−0.059	0.21	0.078	0.94	0.62 to 1.42	0.77
Baseline plaque volume by quartiles	0.83	0.20	15.95	2.29	1.52 to 3.44	<0.001
Lipid-lowering treatment intensity	0.66	0.28	5.49	1.93	1.11 to 3.36	0.02
Baseline PCAT_RCA_	−0.025	0.017	2.01	0.97	0.94 to 1.01	0.15
**B. Progression of high-risk plaque features**	Coefficient	Standard error	Wald	Odds ratio	95% Cl	*p*-values
Age	0.008	0.025	0.11	1.01	0.96 to 1.05	0.82
Total number of CV risk factors	−0.11	0.22	0.24	0.89	0.57 to 1.39	0.57
Baseline plaque volume by quartiles	−0.005	0.23	0.0006	0.99	0.62 to 1.58	0.68
Lipid-lowering treatment intensity	0.28	0.35	0.64	1.33	0.66 to 2.68	0.42
Baseline PCAT_RCA_	0.028	0.012	4.80	1.03	1.00 to 1.05	0.03

CV indicates cardiovascular; RCA, right coronary artery; and PCAT, pericoronary adipose tissue.

## Data Availability

Any underlying research materials related to this paper (for example data, samples, or models) can be accessed upon request to the corresponding authors.

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
