# Peer review of "Longitudinal Effects of Lipid-Lowering Treatment on High-Risk Plaque Features and Pericoronary Adipose Tissue Attenuation Using Serial Coronary Computed Tomography"

_diagnostics, 2025, doi:10.3390/diagnostics15182340_

Round 1

Reviewer 1 Report

Comments and Suggestions for Authors

The study being examined offers a detailed multi-centre observational analysis of the effects of varying intensities of lipid-lowering therapy on high-risk coronary plaque characteristics and the attenuation of pericoronary adipose tissue (PCAT), evaluated via serial coronary computed tomography angiography (CCTA). The authors fill an important gap in the current literature by moving beyond the traditional binary classification of statin therapy and instead breaking it down into low, moderate, and high-intensity regimens. This method enables a more nuanced comprehension of how treatment intensity influences the natural progression of high-risk plaque characteristics. The examination of PCAT, an imaging biomarker gaining recognition for its prognostic significance, provides additional insight by investigating its correlation with both plaque morphology and treatment outcomes.

The study's methodology is enhanced by a substantial patient cohort (n=216) sourced from eleven imaging centers, employing stringent inclusion criteria centered on patients with stable coronary artery disease. Imaging protocols were well coordinated, with all cases analyzed utilizing equipment from a single vendor and, in more than 80% of instances, the same scanner model. Plaque analysis was performed using validated semi-automated software, guaranteeing reproducibility and reducing observer bias. The plaque feature score (PFS) offered a structured, semi-quantitative approach to assessing plaque vulnerability, extending beyond mere volumetric changes to include morphological characteristics linked to acute coronary events.

The results are important for medical practice. High-intensity lipid-lowering therapy was associated with a significantly increased probability of regression in high-risk plaque characteristics approximately four times greater than low or no therapy even after controlling for age, baseline plaque burden, and cardiovascular risk factors. This regression was closely linked to a decrease in the volume of non-calcified plaque, which backs up the idea of "plaque delipidisation" as a way to keep things stable. It is also important to note that finding baseline PCAT attenuation around the right coronary artery can predict plaque progression, regardless of other risk factors or treatment intensity. This supports the idea that PCAT could be a stand-in for vascular inflammation that can be used in risk stratification.

The research has some flaws. Because it was designed to look back in time, it is hard to draw causal conclusions. Also, the fact that different types of scanners were used, even though they were rare, may have caused small differences in measurements that are especially important for PCAT analysis. PCAT measurements were accessible in merely 70% of participants, thereby limiting the robustness of associated findings. Because there isn't any long-term clinical outcome data, we have to guess what the imaging changes mean for this group based on what we've seen in other studies.

From a clinical standpoint, the study corroborates existing guideline recommendations that endorse high-intensity lipid-lowering interventions for patients with heightened cardiovascular risk. The study reinforces the case for vigorous lipid modification as a component of plaque stabilization by showing a graded, treatment-intensity–dependent effect on high-risk plaque regression. Interestingly, the absence of a direct correlation between alterations in PCAT and plaque regression implies that the anti-inflammatory effects of lipid-lowering therapy may be facilitated by mechanisms not entirely represented by PCAT metrics, highlighting the intricacy of vascular biology.

Subsequent research would be enhanced by prospective, randomized designs that include standardized imaging intervals, stringent adherence monitoring, and the incorporation of biochemical markers of inflammation in conjunction with imaging endpoints. Connecting these changes to hard cardiovascular outcomes would give us the definitive clinical proof that observational designs can't fully provide. The moderate interobserver agreement in high-risk plaque assessment underscores the necessity for enhanced training or more sophisticated automated detection techniques to guarantee uniform evaluation in both research and clinical settings.

Reviewer 2 Report

Comments and Suggestions for Authors

This multicenter retrospective study included 216 patients with suspected or established CAD who underwent two serial CCTA examinations (median follow-up ~825 days). Changes in high-risk plaque features (HRPF) and pericoronary adipose tissue attenuation (PCAT, HU) were compared across lipid-lowering intensity (none/low, moderate, high). A semi-quantitative Plaque Feature Score (PFS) summarized HRPF progression/regression, and semi-automated software quantified plaque volumes. Main findings: high-intensity therapy was more likely to be associated with HRPF regression than lower intensity (HR ~4.6), and regression tracked with reductions in non-calcified plaque volume (“delipidization”). PCAT attenuation decreased with higher treatment intensity overall, but PCAT changes were not significantly correlated with PFS/volume changes; baseline RCA PCAT predicted HRPF progression. The following are detailed review comments:

1. The high-intensity group had greater baseline CAD burden, more prior events, and larger plaque volumes, indicating confounding by indication. We recommend propensity-score methods (matching or stabilized IPTW) and/or multilevel modeling, incorporating age, sex, risk factors, CAD-RADS, baseline PFS, baseline plaque volumes, prior MI/PCI, and center/scanner parameters. If feasible, report an E-value to gauge the potential impact of unmeasured confounding.

2. Patients receiving PCSK9 inhibitor therapy were classified as high intensity, but LDL-C goal attainment, on-treatment lipid changes, and time-varying exposure/adherence are not described (baseline lipids available for approximately half). This weakens causal interpretation. Please add analyses of on-treatment LDL-C and absolute reductions, an intensity × time metric (e.g., statin-intensity years), a PCSK9-treated subgroup, and document medication adjustments/discontinuations.

3. Including only patients who completed repeat CCTA may exclude higher-risk or non-adherent individuals, introducing selection bias.

4. Summing HRPF types per plaque (0–3) and aggregating to the patient level conflates plaque number with feature severity, potentially biasing ΔPFS. Consider plaque-level hierarchical models and/or validated continuous metrics (e.g., low-attenuation plaque volume fraction) as sensitivity analyses.

5. The HRPF kappa (≈0.61) reflects moderate-to-substantial agreement, yet residual variability is non-trivial; consider double-reading with adjudication and report PCAT repeatability.

6. PCAT was measurable in ~68% due to vendor/format constraints, risking informative missingness. Please assess the missing-data mechanism, consider multiple imputation, and repeat key analyses in homogeneous scanner/kVp subsets.

7. With only two time points, the timing of regression/progression is interval-censored. Using Cox models warrants caution. Prefer modeling ΔPFS as a continuous/ordinal outcome with mixed-effects, or use discrete-time/interval-censored approaches; include random effects for patient and center if applicable.

8. In Table 3, the HR for baseline plaque volume is reported per 1 mm³ (HR ~1.002), implying ~0.2% increase in hazard per 1 mm³, which is negligible relative to median volumes (hundreds of mm³). Please rescale to per 100–500 mm³ and provide a distribution–effect plot.

9. Please replace causal wording (“leads to/regression caused by/delipidization leads to”) with associative language (“associated with/consistent with”) given residual confounding and the retrospective design.

10. Baseline PCAT predicting HRPF progression supports an inflammatory phenotype signal, whereas lack of association for changes in PCAT may reflect limited power, measurement error, or asynchronous effects of lipid vs. inflammatory pathways. We suggest a more neutral interpretation and propose prospective mechanistic studies with parallel inflammatory biomarkers.

11. The study evaluates imaging surrogates, not clinical outcomes (e.g., MACE). Please avoid over-extrapolation to prognosis and explicitly state this limitation; do not import prognostic inferences from prior meta-analyses directly to this cohort.

Reviewer 3 Report

Comments and Suggestions for Authors

In this retrospective, multicenter study in 216 patients, serial coronary CT scans are evaluated for the presence and number calcified and non-calcified plaques and high risk plaque features, at baseline and after a quite variable observation and treatment period (463.0-1323.0 days) with lipid-lowering drugs, in addition to other medication. PCAT volumes were also measured.

The authors report that high-intensity lipid-lowering is associated with a significant reduction of high-risk plaque features and attenuation of plaque progression; PCAT volumes also decreased, but no association with calcified plaque volume was observed.

Specific comments:

  1. Abstract: please mention not only the mean age, but also the % of female patients (or male) and the mean duration of lipid-lowering treatment and observation period. From the text, it is not unclear if the time of treatment and observation are the same, please also see one of my comments below.
  2. Also, and as the study is retrospective in nature, any conclusions / speculations about potential outcome benefits should be avoided, especially in the abstract.
  3. The criteria for the initiation of low, medium or high intensity treatment should be stated in the Methods or maybe even in the Results section. The baseline lipid parameters do not seem to be important for this decision, as they were already lower in the high-intensity group.
  4. Table 1 says baseline medication, which includes already lipid-lowering therapy. Text says that treatment was started after the CT scan. Clarify the start of the lipid-lowering therapy.
  5. How was the feature “Number and type of high-risk plaque features by coronary artery territory and time-point.” (data shown in Table 2) at baseline distributed over the patient groups according to the treatment intensity? Did the regression depend on the treatment or rather the presence of a high PFS at the start of the observation? If there is no plaque at the start, any intensity of treatment will not lead to a regression.
  6. The definition and calculation of “plaque de-lipidisation” invented by the authors needs to be defined and explained in the Methods.
  7. The patients also are on additional medication to treat hypertension or diabetes. Multivariable analysis adjusting for these factors should be performed.
  8. Regarding the observed changes in PCAT volumes: did the body weight change?
  9. Figure 2: Legend, text and abstract state that PCAT volumes decrease in patients on high-intensity lipid-lowering therapy. How do the data shown in Figure 2D fit in here?
  10. Table 3: Authors state that lipid-lowering predicted plaque regression independent of the baseline plaque volume. Where are the data supporting this conclusion shown? In Table 2, both parameter are significantly altered. I do not see any analysis connecting both.
  11. Also Table 3: did the regression correlate with the duration of treatment? This parameter is quite variable between the groups.
  12. Was any association between plaque lipid volume (and also PCAT volume) and plasma lipid levels levels observed?

Reviewer 4 Report

Comments and Suggestions for Authors

The manuscript addresses an interesting clinical issue. There are several comments to improve the manuscript.

Please pay attention to the inaccurate capitalization in the title.

Patients who had repeat CCTA presents a stable cohort. Sensitivity analysis comparing included vs. excluded patients regarding the baseline characteristics will assess selection bias.

PCAT data are missing, introducing bias. Address this in detail of perform sensitivity analysis.

Use months for follow up duration.

Wide variability in follow-up may influence plaque progression or regression. Stratify the analysis by follow up duration (< or < 1 y).

Adjust for the scanner type in multivariable analysis.

The authors assume continued treatment but there is no adherence data.

How did you handle missing data?

The number of predictors is large compared to event number. This lead to model overfitting. Use variable selection method to include important variables only and avoid overfitting.

Interobserver agreement for high-risk plaque features was moderate. Discuss this more and its effect in clinical decision

Antihypertensive and antidiabetic medications were recorded but not adjusted for in plaque progression models.

The effect size of PCAT is small. What is the clinical significance of this effect size.

Indicate that the prognostic utility of PCAT requires further validation.

Define all abbrevations before use.

Minor language editing is required.

Round 2

Reviewer 2 Report

Comments and Suggestions for Authors

I appreciate the authors’ thorough, point-by-point replies and the revisions that improved clarity—especially the respecification of baseline plaque volume by quartiles, the refinement of wording to emphasize associations, and the expansion of the limitations. These updates enhance interpretability and transparency.

That said, several core methodological concerns remain only partially addressed. Propensity score approaches or multilevel/stratified modeling to mitigate confounding by indication were not implemented (citing potential overfitting), interval-censoring inherent to two time points was not evaluated with discrete-time/interval-censored or mixed-effects sensitivity analyses, and PCAT missingness (~70% availability) was not probed for mechanism nor handled with multiple imputation or homogeneous-scanner subsets. Given these unresolved issues, the manuscript is closer to “partially addressed—further revision needed” rather than fully meeting the bar for an acceptable revision. I recommend adding at least one robustness analysis (e.g., PS/IPTW or random-effects model), a sensitivity analysis for interval censoring, and a principled approach to handling missing PCAT data before acceptance.
